# A Real-Time BOD Estimation Method in Wastewater Treatment Process Based on an Optimized Extreme Learning Machine

**Ping Yu [1,2,3], Jie Cao [1,2,3], Veeriah Jegatheesan [4,\*] and Xianjun Du [1,4]**

1   College of Electrical and Information Engineering, Lanzhou University of Technology, Lanzhou 730050, China; yup@lut.cn (P.Y.); caoj@lut.cn (J.C.); duxj@lut.cn (X.D.)
2   Key Laboratory of Gansu Advanced Control for Industrial Processes, Lanzhou University of Technology, Lanzhou 730050, China
3   National Demonstration Center for Experimental Electrical and Control Engineering Education, Lanzhou University of Technology, Lanzhou 730050, China
4   School of Engineering, RMIT University, Melbourne, VIC 3000, Australia
\*   Correspondence: jega.jegatheesan@rmit.edu.au; Tel.: +61-3-9925-0810

**Abstract:** It is difficult to capture the real-time online measurement data for biochemical oxygen demand (BOD) in wastewater treatment processes. An optimized extreme learning machine (ELM) based on an improved cuckoo search algorithm (ICS) is proposed in this paper for the design of soft BOD measurement model. In ICS-ELM, the input weights matrices of the extreme learning machine and the threshold of the hidden layer are encoded as the cuckoo's nest locations. The best input weights matrices and threshold are obtained by using the strong global search ability of improved cuckoo search algorithm. The optimal results can be used to improve the precision of forecasting based on less number of neurons of the hidden layer in ELM. Simulation results show that the soft sensor model has good real-time performance, high prediction accuracy, and stronger generalization performance for BOD measurement of the effluent quality compared to other modeling methods such as back propagation (BP) network in most cases.

**Keywords:** Biochemical oxygen demand (BOD); cuckoo search algorithm (CSA); extreme learning machine (ELM); soft sensor; wastewater treatment process

---

## 1. Introduction

The awareness of environmental protection in the society has been gradually improving due to better education on sustainability, and wastewater treatment has become one of the important research topics in the field of environmental protection. Biochemical oxygen demand (BOD) is one of the major parameters of effluent quality indices of wastewater treatment processes. Real-time and accurate monitoring of BOD is the key factor to improve the automatic control of the performance of wastewater treatment processes. However, due to the strong nonlinear and time-varying characteristics of wastewater treatment processes as well as the capacity of current detection methods [1] and measurement accuracies of the sensors and instruments, it is difficult to achieve accurate real-time and on-line measurement of BOD data; this limits the application of closed-loop control in wastewater treatment processes [2]. Hence, there is a great need on how to measure BOD rapidly and accurately to control the wastewater treatment processes.

Recently, the development of soft sensing technology provides a new way of measuring variables which are not measurable on-line in real-time. Especially, researchers have applied soft sensing technology to model the wastewater treatment processes and have achieved good results.

Zhang et al. [3] considered the inflow (Q) as well as the chemical oxygen demand (COD), pH, the suspended solids (SS) and the total nitrogen (TN) of the influent as the auxiliary variables to model a feed-forward three-layer multiple inputs and single output (MISO) neural network, called adaptive growing and pruning (AGP) network. The parameters of the neural network were trained by back propagation (BP) algorithm. This soft sensor model was used to predict the BOD concentration of effluent in their simulation study. Simple learning algorithm has the greatest advantage as it can speed up the measurement, but unfortunately the prediction accuracy is not sufficient for it to be applied in real-time. Qiao et al. [4] proposed a soft sensor method for the measurement of BOD based on a self-organizing neural network with random weights (SNNRW). It used the output weight vector to calculate the sensitivity of the hidden layer nodes to the residual. Based on the sensitivity analysis, it could remove less sensitive nodes of the hidden layer by itself according to the level of the sensitivity. It can achieve higher prediction accuracy while performing much more complex calculations. Liu [5] proposed an online soft BOD measurement method based on an echo sound network (ESN) algorithm. The weights of the ESN are trained and obtained by online learning method. The range of learning rate of ESN network is analyzed and determined by Lyapunov theory, which ensured the convergence of the algorithm. This algorithm improved the accuracy of prediction and the adaptability of the model, but with high computation load.

The extreme learning machine (ELM) is a new feed-forward neural network learning algorithm which was originally proposed by Professor Huang Guangbin of Nanyang Technological University [6]. It has the advantages of simple training process, higher training speed and strong anti-interference ability [7]. The training time is greatly reduced compared to other networks, as ELM only needs to set the number of hidden layer nodes, and the randomly generated input weights and hidden layer thresholds are no longer adjusted in the training process. Through experiments, Han et al. [8] demonstrated that ELM has higher training speed and better generalization ability than BP neural network and support vector machine (SVM). Therefore, soft sensor technology based on ELM learning algorithm has been widely applied in industrial process measurement [9,10].

Compared to BP neural network learning algorithm, ELM can avoid issues such as easy to get local optimum solution [11], poor performance indices and low learning rate. However, ELM algorithm itself has some shortcomings, such as the random selection of the input weights and hidden layer threshold which in general would lead to a poor stability of the network. In order to improve the prediction accuracy of the algorithm, it is necessary to increase the number of the hidden layer nodes [6], and the increase will inevitably reduce the computing speed of the ELM learning algorithm. Thus, the application of ELM has to overcome this contradicting pair of operational needs. Researchers have studied and improved the above problems of ELM in recent years. Zhu et al. [12] introduced differential evolution (DE) algorithm into ELM to obtain the optimal connecting weights between the input layer and hidden layer, and the optimal threshold of hidden layer. These optimal parameters can improve the stability of the network. Yan et al. [13] proposed a regularized extreme learning machine (algorithm) based on discriminative information (called IELM), which can significantly improve the classification performance and generalization ability of ELM. Kassani et al. [14] proposed an incremental method for sparsifying the ELM using a newly devised indicator driven by the condition number in the ELM design matrix, called sparse pseudoinverse incremental-ELM (SPI-ELM), which exhibits better generalization performance and lower run-time complexity compared to ELM. Although they improved the computational speed, training accuracy and generalization performance of ELM algorithm, the least squares-based ELM algorithm still has the problem of randomness of the parameters, which will affect the stability of the network.

Based on the above evaluation of algorithms that exist, a BOD soft sensor based on an improved extreme learning machine is proposed in this paper. Adverse effects of randomness of extreme learning parameters on prediction results and stability of the network are considered while proposing such soft sensor. The parameters of ELM are coded as the cuckoo nest locations, with the corresponding fitness values of the root mean square error (RMSE) between the actual value and the prediction value,

to obtain the optimal parameters of the ELM by using an improved cuckoo search (ICS) algorithm [15]. The fuzzy rough monotone dependence (FRMD) algorithm proposed by Liang et al. [16] to reduce the dimensionality of the data from the BSM1 simulation model [17,18] is used in Matlab simulation. The reduction data is used as the input of the soft sensor and effluent BOD is used as the output of it.

## 2. Materials and Methods

The following steps should be taken to build the Neural Network model: (i) Obtain data and normalizing them. (ii) Carry out data attribute reduction process using fuzzy rough monotone dependence (FRMD) algorithm (mentioned in Section 2.3.2). This would help to reduce the input layer nodes, which would save the computational cost when the model training process is carried out and make the process more efficient. (iii) Train the model using the final data, until it reaches the fitness function $f$ (Equation (13)). (iv) Verify the model using the test data sets. (v) Employ the model into field trials. The most important thing to build a Neural Network model of a process is to obtain valuable data for training and verification. This would guarantee the correctness and effectiveness of the model. The modeling process can easily be implemented in MATLAB.

Our paper focuses on the following three points: The first one is fuzzy rough monotone dependence (FRMD) algorithm for the data attribute reduction process to reduce the data attribute, and simultaneously reduce the number of input layer nodes of the neural network (NN) model. The second one is using an improved cuckoo search algorithm (ICS) to adjust and optimize the input weights and the hidden layer biases during the training process. This would help to improve the prediction accuracy and the stability of the extreme learning machine (ELM) NN model. The third one is using the proposed model to do a real-time BOD estimation study in wastewater treatment process.

### 2.1. Extreme Learning Machine

For $N$ arbitrarily distinct samples of data $(\mathbf{x}_i, \mathbf{t}_i) \in \mathbf{R}^n \times \mathbf{R}^m$, where $\mathbf{x}_i = [x_{i1}, x_{i2}, \ldots, x_{in}]^T \in \mathbf{R}^n$, $\mathbf{t}_i = [t_{i1}, t_{i2}, \ldots, t_{im}]^T \in \mathbf{R}^m$, and an infinitely differentiable activation function of any finite interval, $g : R \to \mathbf{R}$, standard single hidden layer feed-forward networks (SLFNs) with $L$ hidden nodes are mathematically modeled as

$$y(\mathbf{x}_j) = \sum_{i=1}^{L} \beta_i g(\mathbf{x}_j \cdot \mathbf{w}_i + b_i) = \mathbf{o}_j, j = 1, 2, \ldots, N \tag{1}$$

where $\mathbf{w}_i = [w_{i1}, w_{i2}, \ldots, w_{in}]^T$ is the connecting weight matrices of the $i^{th}$ hidden node and the input nodes; $\beta_i = [\beta_{i1}, \beta_{i2}, \ldots, \beta_{in}]^T$ is the connecting weight matrices of the $i^{th}$ hidden node and the output nodes; and $b_i$ is the threshold of the $i^{th}$ hidden node. $\mathbf{x}_j \cdot \mathbf{w}_i$ is the inner product of $\mathbf{x}_j$ and $\mathbf{w}_i$. The network topology with linear output nodes is shown in Figure 1.

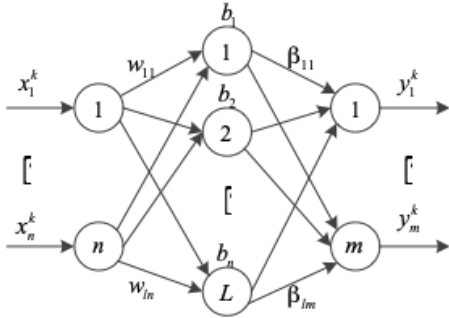

**Figure 1.** Network topology of a standard single hidden layer feed-forward network (SLFN).

Using the initialized random assignment weights $\mathbf{w}_i \in \mathbf{R}^n$ and the thresholds $\mathbf{b}_i \in \mathbf{R}$, the standard SLFNs can approximate these $N$ samples with zero error means that $\sum_{j=1}^{N} \|o_j - t_j\| = 0$, i.e., there exist $\beta_i$, $\mathbf{w}_i$ and $b_i$, such that

$$h(\mathbf{x}_j) = \sum_{i=1}^{L} \beta_i g(\mathbf{x}_j \cdot \mathbf{w}_i + b_i) = \mathbf{t}_j, j = 1, 2, \ldots, N \tag{2}$$

Compact form of Equation (2) can be expressed in matrices as follows:

$$\mathbf{H}\beta = \mathbf{T} \tag{3}$$

where,

$$\mathbf{H} = \begin{bmatrix} h(\mathbf{x}_1) \\ \vdots \\ h(\mathbf{x}_N) \end{bmatrix} = \begin{bmatrix} g(\mathbf{x}_1 \cdot \mathbf{w}_1 + b_1) & \cdots & g(\mathbf{x}_1 \cdot \mathbf{w}_L + b_L) \\ \vdots & \ddots & \vdots \\ g(\mathbf{x}_N \cdot \mathbf{w}_1 + b_1) & \cdots & g(\mathbf{x}_N \cdot \mathbf{w}_L + b_L) \end{bmatrix}_{N \times L} \tag{4}$$

$$\beta = \begin{bmatrix} \beta_1^{\mathrm{T}} \\ \beta_2^{\mathrm{T}} \\ \vdots \\ \beta_L^{\mathrm{T}} \end{bmatrix}_{L \times m} \tag{5}$$

and

$$\mathbf{T} = \begin{bmatrix} \mathbf{t}_1^{\mathrm{T}} \\ \mathbf{t}_2^{\mathrm{T}} \\ \vdots \\ \mathbf{t}_N^{\mathrm{T}} \end{bmatrix}_{N \times m} \tag{6}$$

As named in Huang [19], $\mathbf{H}$ is called the hidden layer output matrix of the neural network; The $i^{\text{th}}$ column of $\mathbf{H}$ is the $i^{\text{th}}$ hidden node output with respect to inputs $x_1, x_2, \ldots, x_N$; The row of matrix $\mathbf{H}$ represents the hidden layer feature mapping with respect to input $x_i$, that is $x_i : h(x_i)$.

If the activation function $g$ is infinitely differentiable and the nodes with parameters of hidden layers can be randomly generated, then there is [6] Theorem 1. Given an small positive value $\varepsilon$ ($\varepsilon > 0$), an activation function g: R→**R** which is infinitely differentiable in any interval and, N arbitrary distinct samples $(x_i, t_i) \in R^n \times R^m$, there exists $L \leq N$ for any parameters of the network $\{(\mathbf{w}_i, b_i)\}_{i=1}^{L}$, according to any continuous probability distribution, then with probability one, $\|\mathbf{H}_{N \times L}\beta_{L \times m} - \mathbf{T}_{N \times m}\| < \varepsilon$.

From the point of view of interpolation, the largest number of hidden layer nodes $L$ should be less than the number of training samples $N$. In fact, when $L$ is equal to $N$, the training error will be zero. According to Theorem 1, when $L$ is less than $N$, SLFNs will approach the training samples with very little training error, and the matrix $\mathbf{H}$ is a non-square matrix, there exists $\hat{\mathbf{w}}_i, \hat{b}_i, \hat{\beta}$, so that Equation (7) can be established.

$$\|\mathbf{H}(\hat{\mathbf{w}}_1, \ldots, \hat{\mathbf{w}}_L, \hat{b}_1, \ldots \hat{b}_L)\hat{\beta} - \mathbf{T}\| = \min_{\mathbf{w}_i, b_i, \beta} \|\mathbf{H}(\mathbf{w}_1, \ldots, \mathbf{w}_L, b_1, \ldots, b_L)\beta - \mathbf{T}\| \tag{7}$$

Unlike the traditional function approximation theories, the input weights $\mathbf{w}_i$ and the hidden layer biases $b_i$ are in fact not necessarily tuned and the hidden layer output matrix $\mathbf{H}$ can actually remain unchanged once random values have been assigned to these parameters in the beginning of learning, and this makes Equation (7) is considerate as a linear system. The training for SLFNs is simply equivalent to finding the least squares solution $\hat{\beta}$ of the linear equations $\mathbf{H}\beta = \mathbf{T}$, that is

$$\|\mathbf{H}(\mathbf{w}_1, \ldots, \mathbf{w}_L, b_1, \ldots b_L)\hat{\beta} - \mathbf{T}\| = \min_{\beta} \|\mathbf{H}(\mathbf{w}_1, \ldots, \mathbf{w}_L, b_1, \ldots, b_L)\beta - \mathbf{T}\| \tag{8}$$

The smallest norm least squares solution of the weights of the above linear system is unique, which is

$$\hat{\beta} = \mathbf{H}^+\mathbf{T} \tag{9}$$

where $\mathbf{H}^+$ is the Moore–Penrose generalized inverse of the matrix $\mathbf{H}$ [20,21].

Thus, the main steps of ELM's learning algorithm can be summarized as follows:

With the given training set $(x_i, t_i) \in R^n \times R^m$, $i = 1, \ldots, N$, the activation function $g(x)$ and the number of the hidden nodes $L$,

**Step 1:** Assign input weight, $\mathbf{w}_i$ and bias of hidden layer, $b_i$, randomly (where $i = 1, \ldots, L$);

**Step 2:** Calculate the hidden layer output matrix, $\mathbf{H}$;

**Step 3:** Calculate the output weight, $\hat{\beta}$.

### 2.2. Improved Cuckoo Search Algorithm-Based ELM (ICS-ELM)

#### 2.2.1. Improved Cuckoo Search (ICS) Algorithm

The cuckoo search (CS) algorithm is approved as an efficient optimization method [22,23]. The principle of CS algorithm is how a cuckoo can find an optimal nest to hatch the eggs by free search based on the obligate brood parasitism and Lévy flights mechanism which is unique in nature. An important advantage of this algorithm is its simplicity. In fact, comparing with other population- or agent-based metaheuristic algorithms such as particle swarm optimization (PSO) and harmony search, there is essentially only a single parameter $p_a$, which represents the probability to be found by the host, in CS (apart from the population size $n$). Therefore, it is very easy to implement.

Its few parameters, simple operation, easy to realize, and good ability to search random paths are the main advantages of CS algorithm. Therefore, once the CS algorithm was proposed, it has been rapidly developed and applied to solve a variety of optimization problems. But at the same time, clearly, CS has some disadvantages such as slow convergence speed and lack of adaptability. We proposed an improved cuckoo search algorithm, named improved cuckoo search (ICS) algorithm [15], which introduces the cosine cyclic operator into the CS algorithm to realize the periodic change of $p_a$ and an adaptive dynamic adjustment strategy for the search step size $S$, which are described as follows.

$$p_a(t) = p_{a,\max}\left|\cos\left(\frac{2\pi}{T}t\right)\right| + p_{a,\min} \tag{10}$$

$$S = \frac{m}{bestX_{i-1}} \times \exp\left(-k \times \left(\frac{t}{t_{\max}}\right)^p\right) + S_{\min} \tag{11}$$

where, in Equation (10), $T$ is the cycle of the periodic operator; $t$ is the evolution generation of the current iteration; $p_{a,\max}$ and $p_{a,\min}$ are the dynamic control parameters of $p_a$ which are equal to 0.75 and 0.1 respectively. In Equation (11), $m \in (0,1)$ is a regulatory factor; $bestX_{i-1}$ is the optimal nest position of the last generation groups; is the limiting factor; $t$ and $t_{\max}$ are the current iteration number and the maximum iteration number; $S_{\min}$ is the minimum search step; $p$ is an integer from 1 to 30.

Based on the above analysis and improvements, the ICS algorithm steps can be described as follows:

Step 1: Set the objective function and initialization function; generate initial population of $n$ host nests $x_i$ ($i = 1, 2, \ldots, n$); set the size of population, the dimension of independent variables, the maximum iteration number, the maximum and minimum probability of being detected.

Step 2: Calculate the current optimal nest position by putting $x_i$ into the objective function.

Step 3: Record the location of the nest, and use Equation (11) to calculate the current step size $S$, then use Equation (12) to update the location of the nest.

$$x_i^{t+1} = x_i^t + \alpha \oplus S, \ i = 1, 2, \ldots, n \tag{12}$$

where, the product of $\oplus$ is a kind of calculation means entrywise multiplications; $\alpha > 0$ is a step size control factor.

Step 4: Compare the current value of the objective function with the last value; Update the value if the function value is better than the previous one; otherwise, keep it unchanged.

Step 5: After updating the nest location, choose a random number $\varepsilon \in [0, 1]$, which obeys a uniform distribution; if $\varepsilon > p_a$, randomly change the value of $x_i^{t+1}$; otherwise leave as it is. Keep the optimal nest position at last.

Step 6: Return to Step 2 if the iteration number has not reached the maximum iteration number; otherwise, continue to the next step.

Step 7: Output the optimal nest location.

You can find more details of the ICS algorithm in Du et al. [15].

### 2.2.2. ICS-Based ELM

When the structure of ELM has been fixed, the network needs to be trained offline first before using the soft sensor online. To improve the prediction accuracy, the input weights $\mathbf{w}_i$ and the hidden layer biases $b_i$ should be optimized through the training process by ICS algorithm.

The root mean square error (RMSE) of the actual value and the predicted value is taken as the fitness value of the nest of each group, shows in Equation (13).

$$f(\boldsymbol{\alpha}_i) = \sqrt{\frac{\sum\limits_{j=1}^{N_{train}} \| \sum\limits_{i=1}^{m} \boldsymbol{\beta}_i g(\mathbf{w}_i \times \mathbf{x}_j + b_i) - \mathbf{t}_j \|_2^2}{m N_{train}}} \tag{13}$$

where, $N_{train}$ is the number of training samples; $m$ is the number of the hidden nodes; $f$ is the fitness function.

### 2.3. Experimental Data Processing

#### 2.3.1. Acquisition of Experimental Data from Benchmark Simulation Model No. 1 (BSM1)

The experimental data used in our research are obtained from the outputs of a 14-day simulation on the benchmark simulation model No. 1 (BSM1) [17] developed by International Water Association (IWA). Data set provided by the European Co-operation in the field of Scientific and Technical Research (COST) based on the actual water quality of the influent of the wastewater treatment plant was used as the simulation input for the experimental process. The data represented four conditions namely steady state as well as dry, rainy, and stormy weathers. The plant consists of 5 bioreactors and a ten-layer secondary settler which is shown in Figure 2.

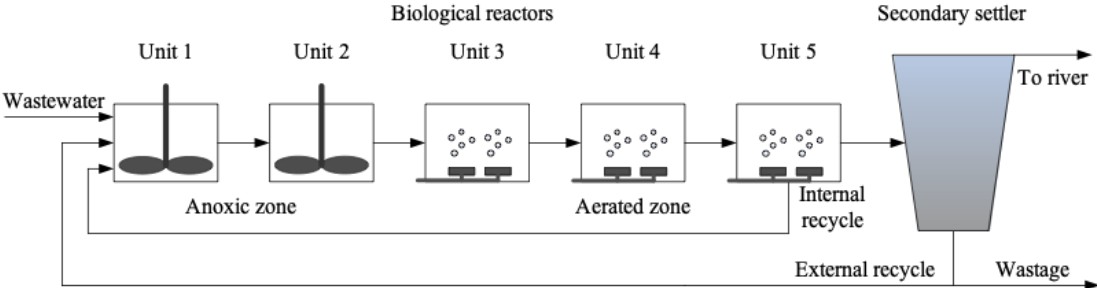

**Figure 2.** Schematic representation of the benchmark simulation model No. 1 (BSM1).

The initial conditions of the operating parameters of the main executor in BSM1 are set as follows (Table 1):

**Table 1.** Parameters of BSM1.

| Parameters | Values | Units | Descriptions |
|---|---|---|---|
| $KLa_3$, $KLa_4$ | 240 | mg/day | Oxygen transfer coefficient of the 3rd and 4th bioreactors |
| $KLa_5$ | 83 | mg/day | Oxygen transfer coefficient of the 5th bioreactor |
| $Q$int | 55338 | m$^3$/day | Internal recirculation flow rate |
| $Q$r | 18446 | m$^3$/day | Returned sludge flow rate |
| $Q$w | 385 | m$^3$/day | Waste sludge flow rate |

Steps to obtain the experimental data:

Step 1: Run a 100-day steady state simulation and repeat until the system achieves steady stability;

Step 2: Run a 14-day simulation with the input data representing dry weather conditions and repeat until the system achieves dynamic stability;

Step 3: Simulate BSM1 to obtain experimental data using the influent data mentioned above.

The 2-week data (1344 groups in total with a sampling time of 15 minutes, 4 groups × 24 hours × 14 days = 1344 groups), of the wastewater treatment process was finally obtained. The compositions of wastewater are shown in Table 2.

**Table 2.** Components of wastewater. COD: chemical oxygen demand.

| Component | Unit | Description |
|---|---|---|
| $S_i$ | mg COD/L | Soluble inert organic matter |
| $S_s$ | mg COD/L | Readily biodegradable substrate |
| $X_i$ | mg COD /L | Particulate inert organic matter |
| $X_s$ | mg COD/L | Slowly biodegradable substrate |
| $X_{bh}$ | mg COD /L | Active heterotrophic biomass |
| $X_{ba}$ | mg COD/L | Active autotrophic biomass |
| $X_p$ | mg COD /L | Particulate product arising from biomass decay |
| $S_o$ | mg -COD/L | Oxygen (negative COD) |
| $S_{no}$ | mg N/L | Nitrate and nitrite nitrogen |
| $S_{nh}$ | mg N/L | $NH_4^+$ and $NH_3$ nitrogen |
| $S_{nd}$ | mg N/L | Soluble biodegradable organic nitrogen |
| $X_{nd}$ | mg N/L | Particulate biodegradable organic nitrogen |
| $S_{alk}$ | mole/m$^3$ | Alkalinity |
| $TSS$ | mg $S_S$/L | Total amount of solids |
| $Q$ | m$^3$/day | Influent flow rate |

### 2.3.2. Fuzzy Rough Monotone Dependence Algorithm for Data Processing

Obviously, the reaction mechanisms of activated sludge process is very complex, and the process parameters involved are numerous, and therefore the dimension of the obtained wastewater data is too high. This will not only lead to the occurrence of over fitting, but also cause the dimension disaster. In order to avoid these problems and not to affect the accuracy of model prediction, it is necessary to find out the parameters which have great influence on the BOD of the effluent. Therefore, the fuzzy rough monotone dependence (FRMD) [16,24] algorithm is used for processing the data to reduce the attribute.

It can be shown that there is a monotonic dependence between conditional attributes and decision attributes. Based on the FRMD algorithm, the steps of attribute reduction for wastewater data are as follows:

Step 1: Define and initialize a two-dimensional array $D[n,m]$ for the decision table, where the $m^{th}$ column is the decision attribute (that is, the data of effluent water quality), and $1^{st}$ to the $(m-1)^{th}$ columns are the conditional attributes (that is, the data of the wastewater influent);

Step 2: Arrange the decision attribute values in ascending order, and exchange the rows of the conditional attributes corresponding to the ordered decision attribute;

Step 3: Make a circular study of the fuzzy rough monotone dependence relation between each condition attribute value and decision attribute value; Obtain the membership function values;

Step 4: If the membership degree function in the set is monotonically increasing, output is the maximum value of them; otherwise, output is 0.

After the attribute reduction, if the membership degree is 0, the conditional attribute will be discarded. The remaining conditional attributes are considered as influential to the BOD, and will be used as the input of the soft sensor to predict BOD.

## 3. Results and Discussion

### 3.1. Data Attribute Reduction

Using fuzzy rough monotone dependence (FRMD) algorithm to do the data attribute reduction process, BOD is taken as the decision attribute, and components in Table 2 are taken as the conditional attributes. The membership function degrees of each conditional attribute to the BOD are shown in Figure 2. Among them, conditional attributes whose membership degrees are equal to 0 have been discarded.

Figure 3 shows the membership degrees between BOD and the components parameters; the conditional attributes which have zero degree are not shown in Figure 3. Nine attributes are shown in Figure 3, but we can clearly see that the degrees of $X$nd and *TSS* are much closer to zero than others. Therefore, for reducing the complexity of the system, those two attributes were not taken into account in the next simulation. The remaining seven attributes were taken as the inputs of the ELM soft sensor.

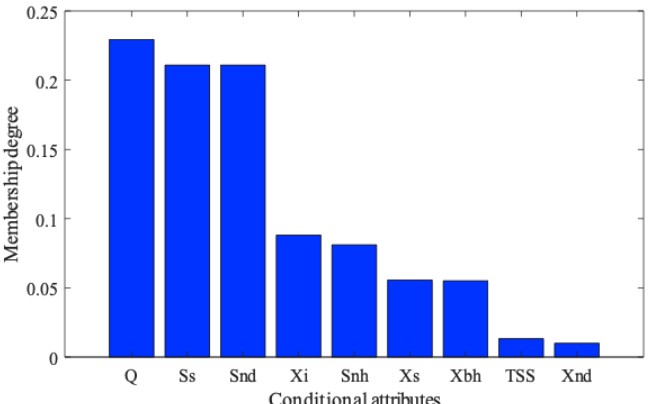

**Figure 3.** Membership degrees of each conditional attribute to biochemical oxygen demand (BOD).

### 3.2. Comparison and Discussion of Simulation Results

The effectiveness of the ICS-based ELM BOD soft sensing model was verified based on the data attribute reduction results; $S_s$, $X_i$, $X_s$, $X_{bh}$, $S_{nh}$, and $S_{nd}$ of the influent and the flow rate $Q$ are used as the auxiliary variables for the network input, so the ICS-based ELM network structure had 7 input nodes, 40 hidden nodes and one output node (7-40-1). The 1344 groups of data are randomly divided into training datasets (500 groups), verifying datasets (460 groups), and testing datasets (384 groups) for the simulation. The training accuracy and prediction accuracy of the soft sensor model are represented by mean square error (MSE).

$$MSE = \frac{1}{N}\sum_{i=1}^{N}\left(Y(i) - Y^*(i)\right)^2 \tag{14}$$

where, $Y(n)$ is the predicted output of the model; $Y^*(n)$ is the actual measured value; $N$ is the sample size number.

Parameters are set as follows: For the ICS algorithm, the population size $n$ is 25, the maximum and the minimum probability to be discovered by the host bird are 0.75 and 0.1 respectively; for the adaptive step length control parameters $m = 0.8$, $k = 0.2$, $p = 25$, $S_{\min} = 0.01$.

When the number of iterations $t = 100$, the iteration will be terminated; the curve of the optimization process is shown in Figure 4. The prediction results and errors of ICS-based ELM soft sensor for effluent BOD are shown in Figure 5.

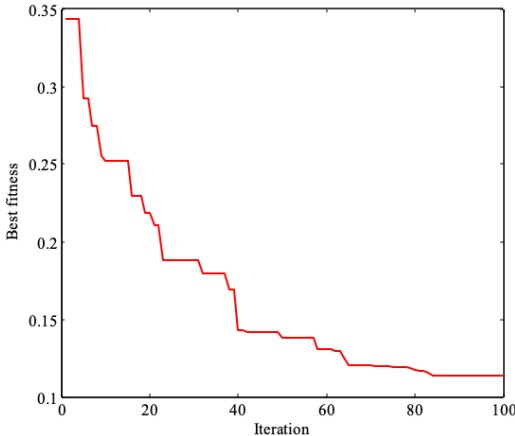

**Figure 4.** Optimization process.

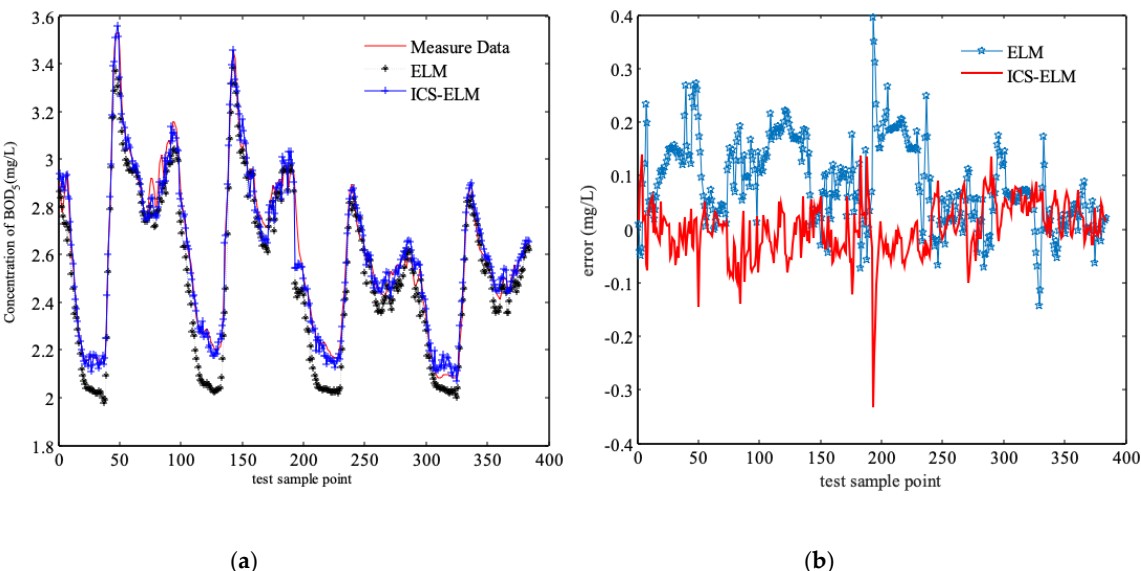

        (**a**)                                         (**b**)

**Figure 5.** Compared prediction results of improved cuckoo search algorithm-based extreme learning machine (ICS-ELM) and basic ELM under dry weather condition, (**a**) effluent BOD concentration, (**b**) error in predicting the measured effluent BOD.

It can be seen from the results, the ICS-based ELM has a better prediction accuracy than the basic ELM. To verify the advantage of the ICS-based ELM, a comparison studies are simulated with the other five models, which are extreme learning machine (ELM), cuckoo search (CS)-based ELM, relevance vector machine (RVM), back-propagation (BP) neural network and least squares support vector machines (LS-SVM), with the same influent data under dry weather condition. The MSE results are shown in Table 3, and the prediction results are shown in Figure 6. Clearly, the MSE from ICS-based ELM are much smaller than the other five models. But the training process will take a slightly longer computer time than for the basic ELM because of the optimization process of the input weights $\mathbf{w}_i$ and the hidden layer biases $b_i$.

**Table 3.** Prediction results of the six soft sensor models. MSE: mean square error; CS-ELM: cuckoo search-extreme learning machine; RVM: relevance vector machine; BP: back propagation; LS-SVM: least squares support vector machines.

| Model | MSE | Hidden Nodes | Training Time (sec) |
|---|---|---|---|
| ELM | 1.3011 | 40 | 1.78 |
| CS-ELM | 0.0640 | 15 | 76.67 |
| RVM | 0.0513 | - | - |
| BP[1] | 0.0909 | 25 | - |
| LS-SVM[1] | 0.0865 | - | - |
| ICS-ELM | 0.0254 | 40 | 61.4 |

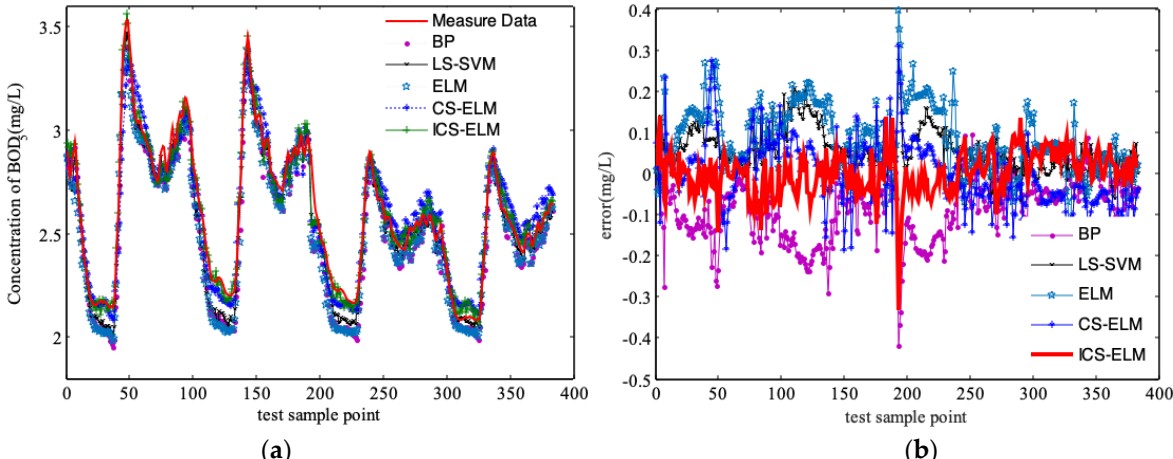

**Figure 6.** Prediction of effluent BOD under dry weather condition using five soft sensors, (**a**) effluent BOD concentration, (**b**) error in predicting the measured effluent BOD.

To further verify the anti-interference ability of the ICS-based ELM prediction model, rainy and stormy weather conditions are considered as the disturbances of the system to simulate the BOD as shown in Figures 7 and 8.

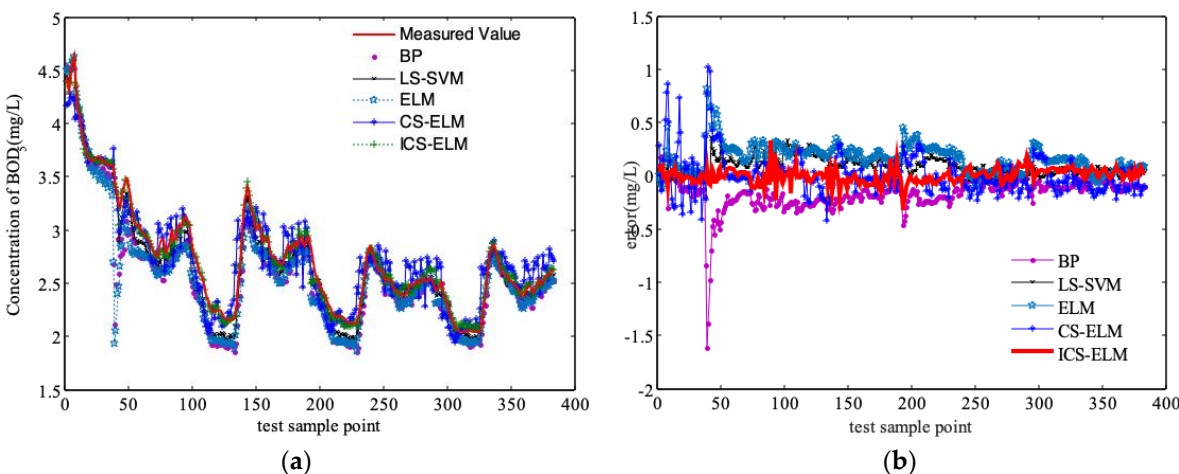

**Figure 7.** Prediction of effluent BOD concentration under rainy condition, (**a**) Effluent BOD concentration, (**b**) Error in predicting the measured effluent BOD.

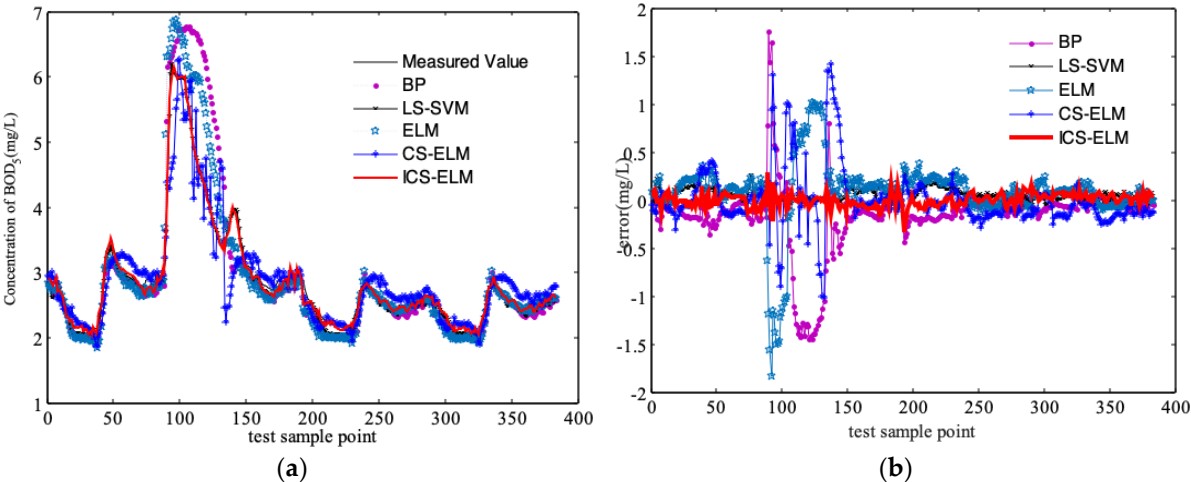

**Figure 8.** Prediction of effluent BOD under stormy weather condition with a storm event, (**a**) Effluent BOD concentration, (**b**) error in predicting the measured effluent BOD.

As can be seen from the results shown in Figures 7 and 8, no matter how the weather condition changes, a well-trained ICS-based ELM still can predict the effluent BOD with smaller errors compared to other five models considered in this paper.

## 4. Conclusions

In this paper, an ICS-based ELM is applied to BOD soft sensing modeling to predict the effluent water quality. It overcomes low prediction accuracy and poor stability of basic ELM algorithm with an improved cuckoo search algorithm. The input weights and the hidden layer biases of ELM are optimized with an offline training process. Results show that ICS-based ELM BOD soft sensing model can improve the accuracy of the prediction with better anti-interference and generalization abilities than basic ELM algorithm. Because of the accurately prediction of the BOD in the process, it would be helpful to the energy saving in the aeration operation in the future research. In summary:

(1) The fuzzy rough monotone dependence (FRMD) algorithm was used to do the data attribute reduction process. This allowed finding out the input parameters closely related to BOD based on the membership function degrees of each conditional attribute. This would help to reduce the input layer nodes, which would save the computational cost when training the model and make the process more efficient.

(2) A periodic change of $p_a$ and an adaptive dynamic adjustment strategy are designed to improve the cuckoo search algorithm. The input weights and the hidden layer biases of the proposed ICS-based ELM are optimized during the offline training process. Through the verification of the results, the proposed method can effectively improve the accuracy of the prediction with better anti-interference and generalization abilities.

(3) Comparing the simulation results of ICS-based ELM, CS-based ELM, basic ELM, LS-SVM, and BP models showed that no matter how the weather condition changes, a well-trained ICS-based ELM still can predict the effluent BOD with smaller errors compared to other five models considered in this paper.

**Author Contributions:** Conceptualization, P.Y., J.C., V.J. and X.D.; methodology, X.D.; software, P.Y.; validation, P.Y., J.C., V.J. and X.D.; formal analysis, P.Y. and X.D.; investigation, X.D.; resources, P.Y.; data curation, J.C.; writing-original draft preparation, P.Y. and X.D.; writing-review and editing, V.J.; visualization, P.Y.; supervision, J.C. and V.J.; project administration, X.D.; funding acquisition, J.C. and X.D.

**Funding:** This research was funded by the National Natural Science Foundation of China, grant number (No. 61563032), the Natural Science Foundation of Gansu Province (No. 1506RJZA104, No. 2017GS10945), University Scientific Research Project of Gansu Province (No. 2015B-030), and the Excellent Young Teacher Project of Lanzhou University of Technology (No. Q201408).

**Conflicts of Interest:** The authors declare no conflict of interest.

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
