# Peer review of "A Real-Time BOD Estimation Method in Wastewater Treatment Process Based on an Optimized Extreme Learning Machine"

_applsci, doi:10.3390/app9030523_

Reviewer 1 Report

This manuscript by Yu et al. aims to assess the potential of an optimised extreme learning machine for the real-time estimation of BOD. In my view this manuscript is well-structured and written. The scope of this article is of high interest for both the scientists and the companies, even some weaknesses of this approach could be better emphasize throughout the manuscript. However, this manuscript deserves for publication in Applied Sciences, and can be accepected as submitted.

Author Response

Thanks and respect for your efficient work and positive comment.

Reviewer 2 Report

Review

This article presents an optimized extreme learning machine (ELM) based on an improved cuckoo search algorithm (ICS) for the design of soft BOD measurement model. The author’s present different steps for the building of this model. Thereafter the author has also presented the application of the obtained model in real case (using of experimental data). This method is always of great interest to area of wastewater treatment. In this respect, the topic of this study is interesting and the approach is valuable in the aspect of practical application. It is well organized. However, it still requires revision before being accepted; the details are listed below.  

Comment :

1.      Materials and methods

-          Please add some information concerning your building of the model. This will greatly help readers understand the process along with the results in this article.

-           

2.      Conclusion

It is short! Try to present all the obtained result and highlight the novelty of current study.

Author Response

Comment :

1.      Materials and methods

-          Please add some information concerning your building of the model. This will greatly help readers understand the process along with the results in this article.

Our response: 

The following is added (red text in the revised manuscript) at the beginning of section 2.

The following steps should be taken to build the Neural Network model: (i) Obtain data and normalizing them. (ii) Carry out data attribute reduction process using fuzzy rough monotone dependence (FRMD) algorithm (mentioned in section 2.3.2). This would help to reduce the input layer nodes, which would save the computational cost when the model training process is carried out and make the process more efficient. (iii) Train the model using the final data, until it reaches the fitness function f (Equation 13). (iv) Verify the model using the test data sets. (v) Employ the model into field trials. The most important thing to build a Neural Network model of a process is to obtain valuable data for training and verification. This would guarantee the correctness and effectiveness of the model. The modeling process can easily be implemented in MATLAB.

 Our paper focuses on the following three points: The first one is fuzzy rough monotone dependence (FRMD) algorithm for the data attribute reduction process to reduce the data attribute, and simultaneously reduce the number of input layer nodes of nural network (NN) model. The second one is using an improved cuckoo search algorithm (ICS) to adjust and optimize the input weights and the hidden layer biases during the training process. This would help to improve the prediction accuracy and the stability of the Extreme learning machine (ELM) NN model. The third one is using the proposed model to do a real-time BOD estimation study in wastewater treatment process.

2.      Conclusion

It is short! Try to present all the obtained result and highlight the novelty of current study.

Our response:

The following is added (red text in the revised manuscript) to highlight the novelty of current study:

In summary:

(1) Fuzzy rough monotone dependence (FRMD) algorithm was used to do the data attribute reduction process. This allowed finding out the input parameters closely related to BOD based on the membership function degrees of each conditional attribute. This would help to reduce the input layer nodes, which would save the computational cost when training the model and make the process more efficient.

(2) A periodic change of pa and an adaptive dynamic adjustment strategy are designed to improve the cuckoo search algorithm. The input weights and the hidden layer biases of the proposed ICS-based ELM are optimized during the offline training process. Through the verification of the results, the proposed method can effectively improve the accuracy of the prediction with better anti-interference and generalization abilities.

(3) Comparing the simulation results of ICS-based ELM, CS-based ELM, basic ELM, LS-SVM and BP models showed that no matter how the weather condition changes, a well-trained ICS-based ELM still can predict the effluent BOD with smaller errors compared to other five models considered in this paper.